

# Early life stages of a Mediterranean coral are vulnerable to ocean warming and acidification

Chloe Carbonne[1], Steeve Comeau[1], Phoebe T.W. Chan[1], Keyla Plichon[1,2], Jean-Pierre Gattuso[1,3], Núria Teixidó[1,4]

[1] Sorbonne Université, CNRS, Laboratoire d'Océanographie de Villefranche, 181 chemin du Lazaret, 06230 Villefranche-sur-mer, France

[2] MSc MARRES, Université Côte d'Azur, Sophia Antipolis Campus, 06103 Nice, France

[3] Institute for Sustainable Development and International Relations, Sciences Po, 27 rue Saint Guillaume, F-75007 Paris, France

[4] Stazione Zoologica Anton Dohrn, Ischia Marine Centre, Department of Integrated Marine Ecology, Punta San Pietro, 80077, Ischia (Naples), Italy

*Correspondence to*: Chloe Carbonne (chloe.carbonne@imev-mer.fr)

**Abstract.** The ability of coral populations to recover from disturbance depends on larval dispersion and recruitment. While ocean warming and acidification effects on adult corals are well documented, information on early life stages is comparatively

scarce. Here, we investigate whether ocean warming and acidification can affect the larval and juvenile development of the Mediterranean azooxanthellate coral *Astroides calycularis*. Larvae and juveniles were raised for 9 months at ambient (23°C) and warm (26°C) temperatures and ambient (8.0) and low pH (7.7, on the total scale). The timing of the larvae metamorphosis, growth of the juvenile polyp, and skeletal characteristics of the 9-month-old polyps were monitored. Settlement and metamorphosis were more successful and hastened under a warm temperature. In contrast, low pH delayed the metamorphosis

and affected growth of the recruits by reducing the calcified area of attachment to the substrate, as well as by diminishing the skeleton volume and the number of septa. However, skeleton density was higher under low pH and ambient temperature. The warm temperature and low pH treatment had a negative impact on the survival, settlement, and growth of recruits. This study provides evidence of the threat that represents ocean warming and acidification for the larval recruitment and the growth of recruits of *A. calycularis*.

**1 Introduction**

Anthropogenic atmospheric $CO_2$ emissions are driving major global threats for corals such as ocean warming and acidification (Kleypas et al., 2001). Under the high-$CO_2$ emission scenario SSP5 - RCP 8.5, sea surface temperature is expected to increase globally by +3.2°C and ocean pH to decrease by 0.3 units by the end of the century (Kwiatkowski et al., 2020). Ocean warming has well-documented negative impacts on tropical reefs where it can induce coral bleaching, which leads to a reduction in

growth, reproduction, recruitment, and high mortality over large spatial scales (McClanahan et al., 2009; Sully et al., 2019). Ocean acidification, the change in carbonate chemistry resulting from the uptake of atmospheric $CO_2$ by the ocean (Gattuso and Hansson, 2011), is responsible for decreasing seawater pH and calcium carbonate saturation state (Orr et al., 2005). Calcifying species such as corals are perceived to be especially threatened by ocean acidification as many studies have shown a decline in adult tropical coral calcification and growth with a pH reduction (Cornwall et al., 2021). Temperate corals, such

as Mediterranean species, have been suffering regular mass mortalities due to the increased intensity, duration and frequency of marine heatwaves (Garrabou et al., 2019). However, temperate corals seem to be more tolerant to ocean acidification than tropical corals as their calcification is rarely affected by low pH (Rodolfo-Metalpa et al., 2011, Carbonne et al., 2021). The





recovery of coral populations depends not only on adult resilience but also on successful sexual reproduction, larval development and recruitment (Bahr et al., 2020).

Sexual reproduction is essential for the dynamics of populations since it maintains genetic diversity and favors adaptation to changing environmental conditions (Bay and Palumbi, 2014). It also facilitates recovery by replenishing populations after disturbances and maintaining resilience of marine communities (Hughes et al., 2019). Effective sexual reproduction is a complex process defined by different life-history factors: (1) gamete production and fertilization leading to high pelagic larval dispersal, (2) habitat selection by recognition of suitable substrate and high recruitment by settlement, (3) and post-settlement
growth and survival (Ritson-Williams et al., 2009, Albright et al., 2011). Although the production of larvae and early life stages are critical processes for rebuilding adult populations, corals in these early stages can be particularly sensitive to ongoing and projected environmental changes (Adjeroud et al., 2016). However, little is known on the response of coral larvae and recruits to ocean warming and acidification.

Ocean warming can be deleterious to coral larvae. For example, in early development, high temperatures can lead to embryonic
abnormality (Woolsey et al., 2013). Furthermore, the survival of larvae and recruits is frequently reduced under ocean warming (Baria et al., 2015, Bahr et al., 2020). Elevated sea surface temperature is commonly related to higher metabolic rates, resulting in faster metamorphosis (Chua et al., 2013) and settlement (Nozawa and Harrison, 2007), increasing larval retention to the native population.

Unlike warming, the response of coral early life stages to ocean acidification is equivocal. In some cases, lower pH leads to
decreasing larval metabolism, metamorphosis, and settlement (Albright 2011, Nakamura et al., 2011). However, Chua and colleagues (2013) have shown that embryonic development and metamorphosis of two coral species is differently affected by elevated $pCO_2$ (low pH). On the other hand, most of the literature reports little impact of acidification on the survival of larvae and recruits (Suwa et al., 2010; Chua et al, 2013). As for adult corals, low pH has a negative impact on post-settlement calcification (Suwa et al., 2010, Varnerin et al., 2020) and induces skeletal deformities in recruits (Foster et al., 2016).

Although studies of the effects of warming and acidification in isolation are few, even less have assessed the combined effects of both stressors in early-life stages. Regarding adult corals, the combined effects of both high temperature and low pH are generally dependent on their intensities. For example, a moderate warming can counteract the impact of acidification ending up with a neutral effect on calcification of adult corals (Kornder et al., 2018). Warming and acidification will have an additive effect (higher than the individual effects) on calcification rates under the RCP 4.5 and a synergistic effect (more than the sum
of the individual effects) under RCP 8.5 for the end of the century (Cornwall et al., 2021). In the early life stages, warming and acidification induce divergent responses with decreased or enhanced calcification (Anlauf et al.2011, Foster et al., 2015). In contrast, studies have reported no effects of combined warming and acidification on larval survival, metamorphosis and settlement (Anlauf et al., 2011, Chua et al., 2013, Foster et al., 2015). However, most of the studies of the impact of both ocean warming and acidification on early life stages have focused on tropical zooxanthellate coral species (e.g., Albright et al., 2011,
Baria et al., 2015), while studies on temperate corals are scarce (Carroselli et al., 2019, Varnerin et al., 2020).

Here, we assess the response of early life stages of the temperate azooxanthellate coral, *Astroides calycularis*, to the combined effects of elevated temperature and low pH. This colonial scleractinian coral is endemic to the Mediterranean Sea and found between 0 and 50 m depth, more commonly in shallow rocky habitats (Zibrowius, 1995). Colonies are gonochoric (Goffredo et al., 2011) and fertilization occurs from April to May, with sperm release coinciding with increasing photoperiod and water
temperature (Goffredo et al., 2011). *A. calycularis* is an internal brooder as egg fertilization takes place in the coelenteron and female broods the embryos until they are fully developed into mature swimming larvae. Larvae are released at the beginning of summer when temperature increases (~23°C, Goffredo et al., 2011). Previous studies on *A. calycularis* adult colonies have



shown that this coral is tolerant to ocean warming and acidification (Movilla et al., 2016, Teixido et al., 2020; Carbonne et al., 2021).

In the present study, we hypothesize that elevated temperature and low pH have an additive or synergistic effect on the development and growth of the larvae and recruits of *A.calycularis*. To test this hypothesis, larvae of *A. calycularis* were exposed in a fully factorial design to ambient and warm temperatures (23 and 26°C) and ambient and low pH ($pH_T$ 8.05 and 7.7). They were maintained in the laboratory for 9 months to study the impact of these conditions on the development of larval stages, growth of juvenile polyps, and skeleton density of the 9-month-old polyps.

## 85 2 Materials and Methods

### 2.1 Sampling site and larval release

Fifty-six colonies of *Astroides calycularis* (~5 cm of diameter) were collected on the 1[st] of July 2020 by scuba diving in Ischia, Italy, at the site Sant'Angelo (40°41'31.1"N 13°53'35.0"E, Fig. S1). They were maintained in a 30-liter tank with water motion provided by a NEWA mini 606 pump and maintained in the dark. Larvae were observed by transparency in the gastrovascular
cavity and tentacles of the female colonies (Fig. S2). Larvae were released from the mouth of female polyps when the body contracted or when the colonies were touched (Video S1 and Fig. 1A). The release took place on the night of 1[st]-2[nd] of July 2020; in situ seawater temperature was around 23°C. Released larvae were fully mature (Video S2). They were collected with a pipette, pooled, stored into two 300 mL airtight plastic boxes filled with seawater, and transported to the aquarium facilities of the Laboratory of Villefranche-sur-Mer, France, in less than 12h.

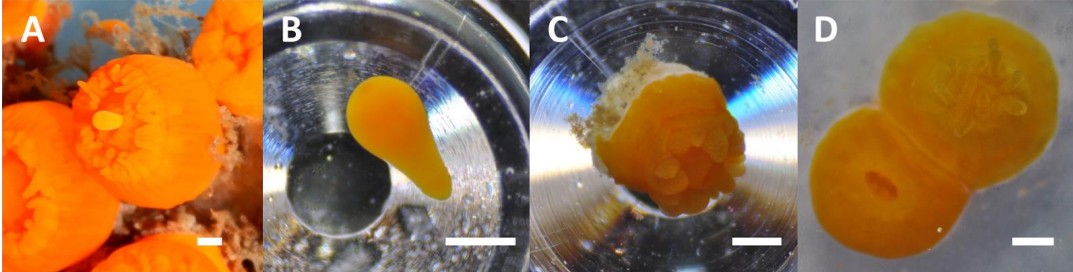


Figure 1. **Early life stages of *Astroides calycularis*** A) spawning of a larva from the mouth of a female polyp, B) a larva after 15 days, C) not-settled metamorphosed polyp after 1 week, and D) settled metamorphosed polyp with 2 polyps after 1 week. Scale: 1 mm

### 2.2 Experimental setup and treatments

After 72 h of acclimation to the laboratory at ambient pH ($pH_T$ = 8.05) and temperature (23°C), the 180 larvae were maintained
in the laboratory under 2 constant pH and 2 constant temperature treatments for 9 months. pH treatments were "*ambient pH*" ($pH_T$ ~ 8.05), and a "*low pH*" ($pH_T$ ~ 7.70), corresponding to a decline of pH projected by the end of the century under the RCP 8.5 $CO_2$ emissions scenario (Kwiatkowski et al., 2020). The two temperature treatments were "*ambient temperature*" which was the in situ temperature at the time of sampling (23°C) and "*warm temperature*" ~26°C, which was shown to induce thermal stress of benthic species in the Mediterranean Sea (Gómez-Gras et al., 2019). The larvae were divided into 12 x 300
mL crystallizers (Duran®), as triplicates with 15 larvae for each of the four treatment conditions (2 pH x 2 temperatures). Each crystallizer was submerged in an independent 5 L experimental tank. While larvae were still swimming, water was changed every day and crystallizers were covered with parafilm® to prevent changes in pH and the escape of larvae. When all larvae



metamorphosed and settled, the crystallizers were covered with a 45 µm plankton net to allow water exchange with the experimental tank. The experimental tanks were gravity-fed (50 ml min$^{-1}$) from six 25 L header tanks (3 at each pH treatment)

continuously supplied by seawater pumped from Villefranche Bay at 5 m depth. pH was controlled using a pH controller (APEX, Neptune Systems) which regulated the delivery of pure $CO_2$. Temperature was regulated in the experimental tanks by temperature controllers (APEX, Neptune Systems) and 300 W ThermoControl-e heater (Eheim), and both pH and temperature were measured every week (see below).

Light was provided by 24W Aquablue Special T5 (ATI). Irradiance was constant from 7 to 19h at 16 µmol photons m$^{-2}$ s$^{-1}$.

Submersible pumps (NEWA) provided water motion in each experimental tank. When juvenile polyps had tentacles (approximately after a month, from the 5th of August 2020), they were fed twice a week with a 10 mL solution of freshly hatched brine shrimps (*Artemia* sp.). 24 h after feeding, the crystallizers were cleaned with a painting brush and the water changed to remove any detritus.

### 2.3 Carbonate chemistry

pH in the header and experimental tanks was measured weekly using a handheld pH-meter (826 pH mobile, Metrohm) calibrated with a seawater pH TRIS buffer (batch #T33 provided by A. Dickson, Scripps Institution of Oceanography, USA) before each set of measurements. Temperature was also measured weekly in each tank with a Traceable™ digital thermometer (FisherBrand). Salinity and total alkalinity ($A_T$) data during the experiment were obtained from the weekly measurements performed in the in the Bay of Villefranche by the *Service d'Observation Rade de Villefranche, SO-Rade*, of the *Institut de la*

*mer de Villefranche and the Service d'Observation en Milieu Littoral, SOMLIT/CNRS-INSU*. This was possible because we worked in an open system where seawater from the Bay of Villefranche was continuously delivered. To confirm that $A_T$ was not altered by metabolic activity, it was measured in each experimental and header tank at the beginning and end of the experiment. It was determined by potentiometric titration using a Metrohm 888 Titrando following the method of Dickson et al. (2007). Titrations of certified reference material (Batch #186) provided by A. Dickson were used to assess the accuracy of

the measurements and were within 6.5 µmol kg$^{-1}$ of the reference value. pH$_T$, temperature, $A_T$ and salinity were used to calculate the other carbonate chemistry parameters using the R package seacarb (Gattuso et al., 2021).

### 2.4 Early life-stages monitoring

The life stages of each 15 individuals (larvae) per crystallizer were recorded every 2 days when swimming larvae were present, and every week after settlement. The different stages were: "planula", when the larvae were still swimming (Fig. 1.B, Video

S3), "settled polyp" when larvae had metamorphosed and settled (Fig. 1.D), "non-settled polyp" when larvae had metamorphosed but not settled or detached from the substrate after settling (Fig. 1.C), and "dead". Missing larvae were assumed dead as coral larvae lyse within 24 h after death (Baird et al., 2006).

Pictures of each larva from each crystallizer were taken every 2 to 3 days with a stereomicroscope (SteREO, Discovery V.12, Zeiss) coupled with a camera (D5100, Nikon) until no swimming planula was present. For each polyp attached to the glass of

the crystallizer, a picture was taken every 2 weeks from below with a camera (Coolpix W300, Nikon) and a ruler used as scale (Fig. 3.A). All pictures were analyzed with ImageJ to measure the maximum length and maximum width of the larvae and the surface of the polyp base (Fig. 3.A). Appearance of new polyps from one initial polyp was recorded and reported as the number of polyps in one colony (Fig. 3.C).



### 2.5 Skeleton analysis

After 9 months, the polyps were collected and placed in 5% sodium hypochlorite for 2 h and then rinsed with MilliQ water to remove organic tissue from the skeleton. The skeletons were dried at ambient temperature for two days. The perpendicular diameters (D and d) and the height (h) were measured using a digital caliper. The volume of the skeleton was calculated using the following equation: $V = \pi * D * d * h$. To count the number of septa, a picture of each skeleton was taken from the top with a stereomicroscope (SteREO, Discovery V.12, Zeiss) mounted with a camera (D5100, Nikon).

Six skeletons of each condition (total of 24) were scanned in a micro-CT scanner (GE Healthcare, eXplore Locus RS, see Micro-CT scanner setup in Supplementary materials, Fig. S3). Micro-CT imaging produced a 3D distribution of linear attenuation coefficients that was stored as an x-ray volume image for each coral specimen, defined by the coral polyp base along the X-Y in-plane axis, and the growth axis following the Z-axis (Fig. 4.A, Video S4). Sample visualization and analysis was performed using MicroView Standard 2.5.0–2702 (Parallax Innovations Inc., 2015) to reconstruct each specimen in 3D,

and to digitize a region of interest (ROI delineating the region to be analyzed). Six coral specimens were imaged in every micro-CT scan, each representing one of the four laboratory treatments (temperature x pH). The ROI was plotted as a square based prism spanning the entire specimen to crop each coral sample out for individual analysis. A threshold greyscale value of 2500 HU was carefully selected to remove pore spaces from coral skeleton. This approach assured consistent sampling focused on skeletal aragonite only. Voxels with a greyscale values below the threshold of 2500 HU were deemed to be empty

space and excluded from the skeletal density calculation. However, small pore spaces (< 20 μm) were included in the skeletal density calculation since they were unresolvable at 20 μm resolution. The skeletal density for each coral polyp ($mg/cm^3$) was determined using the fractional mineral content of each voxel (3D pixel) above the greyscale threshold (to exclude pore spaces), averaged over all the voxels contained within each region of interest, and linearly rescaled to units of pure crystal aragonite (density = 2.95 $g/cm^3$).

### 2.6 Data analysis


Linear mixed-models with a hierarchical structure were used to evaluate the treatment effects through time on the number of larvae in each life stage (planula, settled polyp, non-settled polyp, dead), and the length and width of larvae. Hierarchical linear models were used since data were compiled from repeated measures of the same pool of larvae over time. The models were fitted using the function lmer of the R package lme4 (Bates et al., 2015). The fixed factors of the models were temperature,

pH and time, and crystallizers and time were assigned as random effects. The structure of the random term was selected by comparing models with different error structures using the Akaike information criterion (Table S1). For parameters that are not time-dependent, generalized linear mixed models (GLMM) were used with a Gaussian distribution to test for the effects of the treatments on the surface of the polyps' base, the skeletal density and volume, and a Poisson distribution for the number of polyps per colony and the number of septa. Temperature and pH were fixed factors and crystallizers of each condition a

random factor.

## 3 Results

### 3.1 Experimental conditions

The ambient pH treatment was maintained at a mean $pH_T$ of 8.05 ± 0.09 (mean ± SD, n=324) and the low pH treatment at a mean $pH_T$ of 7.78 ± 0.10 (mean ± SD, n=324, total number of weekly $pH_T$ discrete measures in each ambient pH experimental

tank, Fig. S4.B, Table 1). The *ambient temperature* treatment was maintained at 23.5 ± 1.6 °C (mean ± SD, n=342) and the




*warm temperature* treatment at 26.6 ± 0.8 °C (mean ± SD, n=342, Fig. S4.A, Table 1). Salinity and total alkalinity did not vary much during the duration of the experiment with a salinity of 37.84 ± 0.17 (mean ± SD, n=71, Table 1) and a total alkalinity of 2548 ± 13 µmol kg$^{-1}$ (mean ± SD, n=71, Table 1).

Table 1. **Measured and estimated seawater physiochemical parameters of the two pH treatments in the experimental tanks for salinity (S), temperature (T), total alkalinity (A$_T$), dissolved inorganic carbon (C$_T$), pH$_T$, pCO$_2$, calcite (Ωc) and aragonite (Ωa) saturation.** Values are means ± SD with 25th and 75th percentiles. Calculated concentrations of C$_T$, pCO2, Ωc and Ωa are shown. 1: Parameters measured from discrete water samples. pH and temperature conditions are the experimental pH and temperature treatments.

| pH condition | Temperature condition | pH$_T$ | T (°C) | A$_T$ (µmol kg$^{-1}$) | C$_T$ (µmol kg$^{-1}$) | Salinity | pCO2 (µatm) | Ωc | Ωa |
|---|---|---|---|---|---|---|---|---|---|
| low | warm | 7.77 ± 0.11 n=102 | 26.32 ± 0.87 n=111 | 2548$^1$ ± 13 n=71 | 2350 ± 62 n=102 | 37.84$^1$ ± 0.17 n=71 | 964.42 ± 295.57 n=102 | 3.69 ± 0.83 n=102 | 2.45 ± 0.56 n=102 |
| ambient | ambient | 8.07 ± 0.05 n=102 | 23.23 ± 1.65 n=111 | 2548$^1$ ± 13 n=71 | 2202 ± 42 n=102 | 37.84$^1$ ± 0.17 n=71 | 404.43 ± 59.18 n=102 | 5.83 ± 0.60 n=102 | 3.84 ± 0.41 n=102 |
| low | ambient | 7.76 ± 0.10 n=102 | 23.13 ± 1.70 n=111 | 2548$^1$ ± 13 n=71 | 2375 ± 57 n=102 | 37.84$^1$ ± 0.17 n=71 | 962.62 ± 269.51 n=102 | 3.30 ± 0.73 n=102 | 2.17 ± 0.49 n=102 |
| ambient | warm | 8.04 ± 0.07 n=102 | 26.51 ± 0.76 n=111 | 2548$^1$ ± 13 n=71 | 2188 ± 54 n=102 | 37.84$^1$ ± 0.17 n=71 | 442.88 ± 100.72 n=102 | 6.10 ± 0.78 n=102 | 4.05 ± 0.52 n=102 |

### 3.2 Development of early life stages

Larvae started metamorphosing into settled and non-settled polyps after 5 days under *warm temperature - ambient pH* and the last larva settled after 41 days under *ambient temperature - low pH* condition. The proportion of larvae was significantly affected by the interaction of pH, temperature and time (Fig. 2.A, $F_{1,145}$=4.437, $p$=0.035, Table S2), as larvae under *warm temperature* conditions metamorphosed faster: 50% in 15 days, while *ambient temperature - ambient pH* and *ambient temperature - low pH* conditions reached 50% of metamorphosis after 19 and 25 days, respectively (Fig. 2.A). Settlement of

larvae differed between conditions, only 25% of settlement occurred in the *warm temperature – low pH* condition and more than 40% in the two *ambient pH* treatment conditions (Fig. 2.C) due to a significant interaction among the three factors pH, temperature and time ($F_{1,397}$=29.155, $p$<0.001, Table S2, Fig. 2.C). There was a significant effect of pH, temperature and time on the proportion of metamorphosed but non-settled larvae ($F_{1,397}$=36.369, $p$<0.001, Table S2). The two *low pH* treatment conditions presented an opposite outcome. The *warm temperature – low pH* treatment had 36% of non-settled polyps at the

beginning of the experiment and almost 0% after 3 months (Fig. 2.D). On the other hand, the *ambient temperature - low pH* condition presented a progressive detachment of polyps as the proportion of non-settled polyps increased from 9% after a month until reaching 25% after 6 months (Fig. 2.D). Mortality of larvae and polyps increased for every condition during the first two months of the experiment, and then reached a plateau around 50% (Fig. 2.B). The *warm temperature – low pH* condition presented higher mortality compared to the other conditions, reaching up to 71% with a significant effect of the

combined pH, temperature and time ($F_{1,397}$=110.9, $p$<0.001, Table S2, Fig. 2.B). The treatments did not affect larval size (Fig. S7).





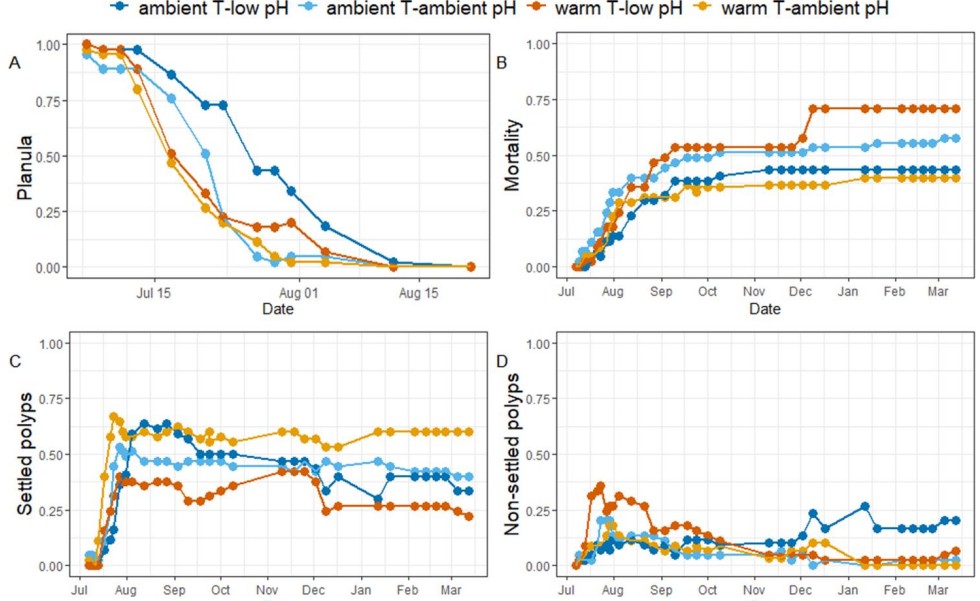

Figure 2. **Proportions of four different stages of *A. calycularis* under the two temperature treatments (23°C for ambient Temperature and 26°C for warm Temperature) and the two pH treatments (pH$_T$ = 8.05 ± 0.09 for ambient pH and pH$_T$ = 7.78 ± 0.10 for low pH) during 9 months.** Panel A) The proportion of planulas (swimming larvae), B) the proportion of mortality (dead larvae and polyps), C) the proportion of settled polyp (metamorphosed and fixed), and D) the proportion of non-settled polyp (metamorphosed but not fixed). The proportion of non-settled polyp includes when larvae metamorphosed into a free polyp and detachment of a settled polyp. n= 180 original larvae, 3 replicates per temperature and pH treatments.

### 3.3 Growth of the recruits

The surface of the polyp bases was significantly affected by pH ($F_{1,35}$= 23.6, Table S3). The base surface in the *ambient pH* treatment were significantly larger than under *low pH*, respectively reaching 32.7 ± 3.6 mm² (mean ± SE, n=18) and 13 ± 2 mm² (n=21) after 9 months of experiment (Fig. 3.B, Fig. S5).



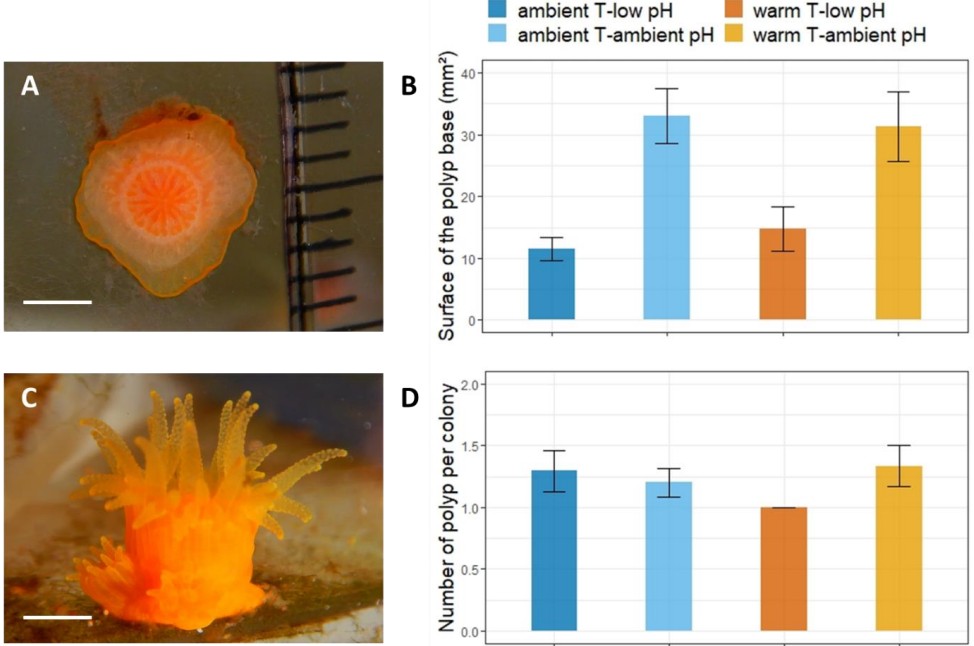

Figure 3. **Mean surface of the recruits' base and mean number of polyps per colony of *A. calycularis* under the two temperature**
**treatments (23°C for ambient Temperature and 26°C for warm Temperature) and the two pH treatments (pH$_T$ = 8.05 ± 0.09 for**
**ambient pH and pH$_T$ = 7.78 ± 0.10 for low pH) after 9 months**. Values are means ± SE. A) Picture of the base of a recruit through the
glass of the crystallizer, B) Mean surface of the recruits' base at 9-month old, C) Picture of a recruit with two new polyps on the outskirt, D)
Mean number of polyps per colony. The color of the bars indicates the temperature and pH treatment. n= 9-20 per condition. Scale : 2 mm.

### 3.4 Development of new polyps

New polyps started to bud on the periphery of the recruits at the end of December 2020 (Fig. S6). The number of polyps per
colony after 9 months of experiment was not significantly different across treatments ($F_{1,55}$= 2.3, *p*=0.135, Table S3). However,
in contrast to all other treatments, recruits in the *warm temperature – low pH* treatment did not develop new polyp during the
9-month of experiment (Fig. 3.D). The recruits in the *warm temperature – ambient pH* treatment exhibited new polyps first as
well as the highest number of polyps per colony at the end of the experiment with a mean of 1.33 polyps per recruit.

### 3.5 Skeleton analysis

The size of the skeleton was significantly impacted by the pH treatment ($F_{1,46}$=4.05, *p*=0.044, Table S3), with a mean volume
of 0.12 cm$^3$ in the *ambient pH* treatment and 0.1 cm$^3$ under *low pH* treatment (Fig. 4.B). The number of septa per skeleton was
significantly lower under low pH ($F_{1,48}$=10.5, *p*=0.002, Table S3). The skeleton had less developed septa (mean ± SE, n=29)
under low pH treatment 8.2 ± 0.6 septa than under ambient pH (10.2 ± 0.6 septa, n=23, Fig. 4.C).



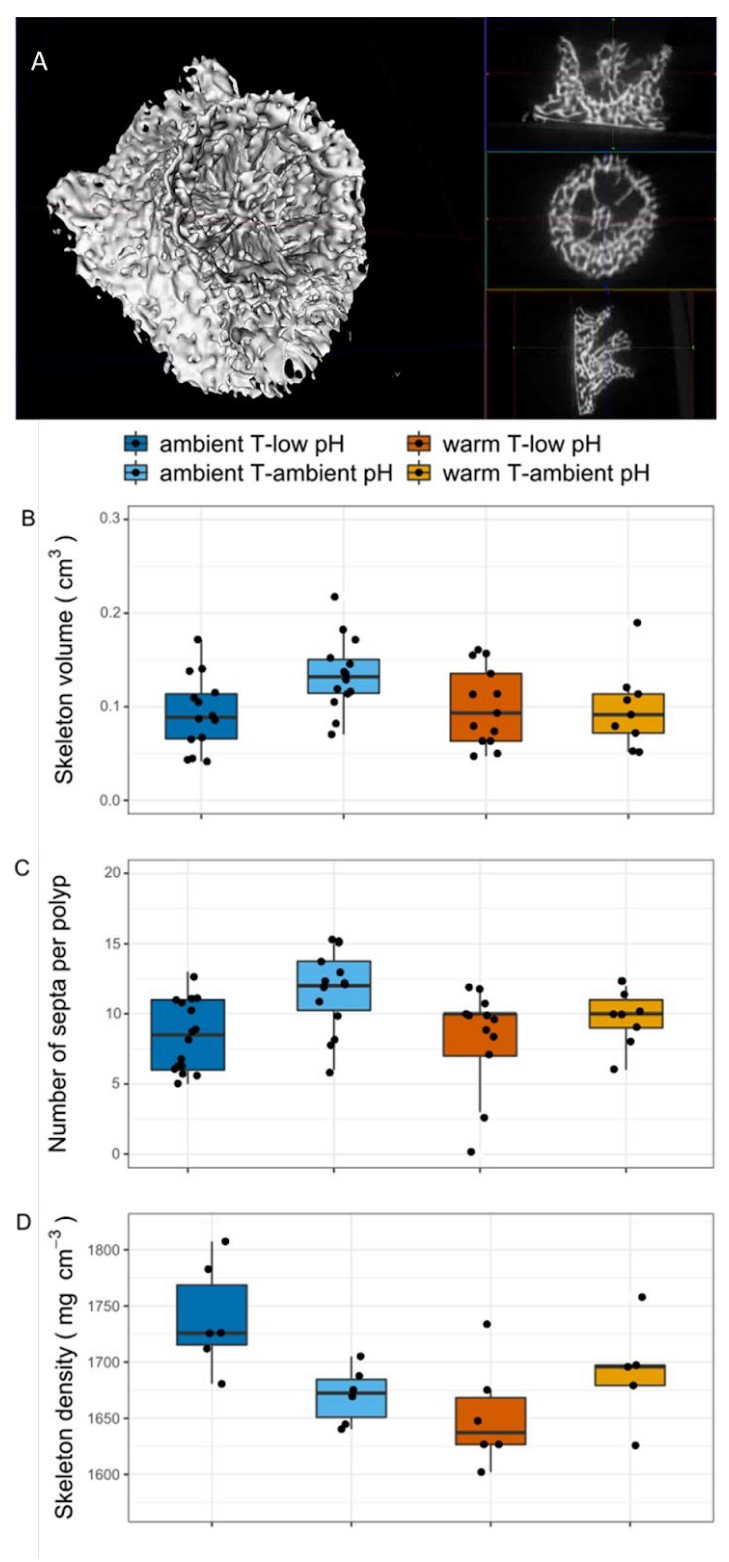






Figure 4. **9 month-juvenile skeletons of *A. calycularis* under the two temperature treatments (23°C for ambient Temperature and 26°C for warm Temperature) and the two pH treatments (pH$_T$ = 8.05 ± 0.09 for ambient pH and pH$_T$ = 7.78 ± 0.10 for low pH).** A) 3D picture of a recruit skeleton by micro CT scanning B) Volume of the recruit. n=9-16 per temperature and pH treatment. C) Number of septa per skeleton. n=9-16 per temperature and pH treatment. D) Skeleton density. Values have been obtained by micro CT scanning. n=6 per temperature and pH treatment. Dots represent the number of septa per recruit coral skeleton and the density of the recruit's coral skeleton and boxes represent median and 25 and 75% quartiles. The color of the boxes indicates the temperature and pH treatment.

The combined effect of pH and temperature had a significant impact on skeletal density (F$_{1,19}$=7.6, p=0.012, Table S3). The skeleton of polyps exposed to *ambient temperature – low pH* condition presented a higher density with 1739 ± 19 mg cm$^{-3}$ (mean ± SE, n=6) than in the other conditions in which it was lower than 1690 mg cm$^{-3}$ (n=6, Fig. 4.A and 4.D)

## 4 Discussion

This experiment shows that settlement and metamorphosis of the larvae are faster under warm temperatures. In contrast, low pH delayed the metamorphosis and affected the skeletal properties of the recruits. The surface of the base of the skeleton and the number of septa per polyp were lower under low pH conditions, whereas the skeleton density was higher. Both drivers combined had a negative synergistic impact on survival and settlement, and an additive impact on growth. Therefore, our results suggest that both warming and acidification, individually or combined, affect the early life stages of *Astroides calycularis*.

Larvae were released by the female colonies at the beginning of summer (end of June), when the sea surface temperatures reached 23°C. Metamorphosis and settlement were accelerated in the warm treatment. Similar results were observed in the majority of similar studies in which larval development was noticeably increased at warmer temperatures (Edmunds et al., 2001, Nozawa and Harrison, 2007, Chua et al., 2013). In contrast, few studies reported a decline in settlement and development of tropical coral larvae under warming (e.g. Randall and Szmant, 2009), possibly due to the high temperature (31°C and 33°C) that reduced larval mobility and increased mortality. Higher temperatures accelerate biochemical processes by increasing the activity of enzymes until a threshold is reached (Clarke and Fraser, 2004). As a result, metabolic rates increase, shortening the duration of the larval phase and reducing the distance of dispersal by causing metamorphosis and settlement before larvae are exported from their population of origin (O'Connor et al., 2007, Figueiredo et al., 2014). Local retention of the larvae has a direct impact on connectivity and genetic diversity, which potentially reduces the resilience and resistance to disturbances and future global change (Bay and Palumbi, 2014). Warmer temperatures are also commonly associated with an increased mortality of early life stages (Baria et al., 2015). Here, the mortality of larvae and recruits was not affected by temperature. This might be because *A. calycularis* is a known thermophilic species, with a South-Western distribution in the Mediterranean Sea. The warm temperature used for the experiment, 26°C, is the mean temperature during the summer months in Ischia (July and August, Teixido et al., 2020). The colonies of *A. calycularis* have been suffering mass mortalities in Ischia and North of Sicily in summer when temperatures reach 28°C (Gambi et al., 2018, Bisanti et al., 2022).

The surface area of the base of the skeleton was highly impacted by acidification, decreasing by 60% under *low pH* compared to the *ambient pH* treatment. The skeletal volume was also reduced by 19% under acidified conditions. Albright and Langdon (2011) also observed a decrease of post-settlement growth of 50% in recruits of *Porites astreoides* with a similar pH$_T$ treatment of 7.7. This difference in growth might be due to a decrease in metabolism under low pH (Albright and Langdon 2011). The formation of the skeleton was also impacted by acidification as the number of septa was less abundant and not presenting a radial symmetry under low pH. This is in agreement with other studies which have reported porous corallite walls, thinner basal plate, and asymmetric skeleton under similar low pH conditions (Foster et al., 2016). This impact on the growth and skeleton structure can likely explain the increased detachment of polyps five months after the beginning of the experiment. A





large number of recruits under low pH had the skeleton base exposed, retracting their tissue towards the upper section of the corallite (Fig. S8). While 17 out of 36 polyps exposed to *low pH* had an exposed skeleton at the base, none of the 33 polyps under *ambient pH* exhibited tissue retraction. The polyps were thus unable to extend their skeleton on the substrate, and the exposed skeleton was likely experiencing dissolution, as described before with *Cladocora caespitosa* by Rodolfo-Metalpa et

al. (2011). This phenomenon can be compared to the loss of coenosarc on adult colonies under acidification observed on colonies of *A. calycularis* found near $CO_2$ seeps (Teixido et al., 2020). Interestingly, while the growth and structure of the skeleton were affected by low pH, the density of the skeleton was higher. These results are in accordance with the higher density observed on adult colonies living in a $CO_2$ vent site in Ischia (Teixido et al., 2020). It suggests that *A. calycularis* responds to low pH by increasing the density of their skeleton, perhaps for greater resistance to mechanical stress, at the

expense of other physiological parameters such as growth and skeleton structure. These results suggest that acidification mainly had repercussions on calcification of the recruits, whereas larval development and survival were not impacted, as observed by Foster et al. (2015).

The proportion of settlement of *A. calycularis* larvae was significantly lower in the warm temperature and low pH treatment. In contrast, other studies reported no effect of such treatment (e.g. Anlauf et al., 2010, Foster et al., 2015). This difference can

be explained by the higher magnitude of the warming and acidification treatments used in the present study compared with other studies (+ 3 vs + 1-2°C and -0.3 vs -0.2 pH unit) (Anlauf et al., 2010, Chua et al., 2013, Bahr et al., 2020). Furthermore, all previous studies focused on zooxanthellate corals. The low settlement rate of *A. calycularis* under the warm and acidified treatment is related to the high number of larvae experiencing metamorphosis with no settlement during the first days of the experiment. In the first 15 days, 36% of the larvae metamorphosed without settling. After metamorphosing, the individuals

floated for a week before sinking, then calcification started and the tentacles appeared. Some of them eventually attached to the bottom after three months. This phenomenon could also explain the high mortality rate observed in the *warm temperature – low pH* condition as non-settled polyps could not feed properly. Recruits were therefore less abundant, and their size was smaller than in any other condition. Warming and acidification presented an additive effect on the surface base of the recruits as the combined effects of warm and acidified waters lead to a 63% reduction in growth, while the low pH treatment caused a

55% decrease and the warm treatment a 6% decrease, compared to the control condition. Anlauf et al., (2010) found a synergistic effect of warming and acidification on the growth of *Porites panamensis* recruits. The size of the recruits was 30% smaller in a combined warm and low pH treatment, while only a 3% decrease was observed under acidification alone and 0% under warming. For *Acropora spicifera* recruits, warming mitigated the impact of acidification as calcification was lower (-60%) under sole acidification than under both warming and acidification (-48%, Foster et al., 2015). The impact of stressors

on calcification is considered species-specific (Comeau et al., 2013). However, methodological differences could also explain the range of responses observed. The duration of our experiment (9 months) was longer than previous ones (6 weeks, Anlauf et al., 2010 and 5 weeks, Foster et al., 2015). For example, in our study, the different responses to the treatments on *A. calycularis* recruits' surface base were only observed 2 months post settlement. Finally, under *warm temperature – low pH* condition, 63% of the recruits presented an exposed skeleton at the base, more than under *ambient temperature - low pH*

condition. However, new polyps of *A. calycularis* bud from the tissue at the base of the recruit periphery. This particular characteristic explains why the recruits under the double stress condition were the only one not presenting new polyps. Thus, warm temperature and low pH together harmed the development of recruits into colonies.

Our findings highlight that warm temperatures and acidification have distinct impacts on the early life stages of *Astroides calycularis*. Temperature acts on larval development while pH acts on the growth and calcification of the recruits. The

combined effect of warming and acidification early life stages will negatively impact the resilience and resistance of the Mediterranean populations by decreasing dispersion, recruitment and post-settlement growth. In order to better predict the



future of *A. calycularis* populations at the end of the century under global change, further research needs to be done to evaluate if acclimatization or adaptation to warming or acidification can occur in early life stages.

**Author contributions**

C.C., S.C., N.T., and J.-P.G designed the study. C.C. and N.T. were involved with fieldwork. C.C., P.C. and K.P. performed the experiments. C.C., N.T. and K.P. analyzed the data. C.C. wrote the first draft of the manuscript which was then finalized by all co-authors.

**Competing interests**

The authors declare that they have no conflict of interest.

**Acknowledgements**

This research was supported by the French Government through the National Research Agency - Investments for the Future ("4Oceans-Make Our Planet Great Again" grant, ANR-17-MOPGA-0001). Thanks are due to the *Service d'Observation Rade de Villefranche* (SO-Rade) of the *Institut de la mer de Villefranche* and the *Service d'Observation en Milieu Littoral* (SOMLIT/CNRS-INSU) for their kind permission to use the Point B data. We kindly thank Christopher J.D. Norley and David

W. Holdsworth from Robarts Research Institute of the University of Western Ontario (Canada) for the use and expertise of their micro-CT scanner. Thanks are also due to the Western University Earth Sciences Department's Dana Minerals Collection for the aragonite crystal used as standard for the micro-CT scanning. We thank Samir Alliouane for assistance in the laboratory, Laura Tamburello for the advices with statistical analysis, and Pietro Sorvino (ANS Diving, Ischia) and Alice Mirasole for assistance in the field.

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
