# Peer review of "Early life stages of a Mediterranean coral are vulnerable to ocean warming and acidification"

_EGUsphere, 2022_

## Author Response (AR1)

Dear Associate Editor,

We are pleased to submit a revised version of our manuscript entitled "Early life stages of a Mediterranean coral are vulnerable to ocean warming and acidification" (ID EGUSPHERE-2022-240) as well as a point-by-point response to the reviewers' comments. We thank reviewer #1 and #2 as well as the Associate Editor for their very kind words and for providing useful comments which have improved the manuscript.

The main request of reviewer #1 was to emphasize the kind of growth measured on our study and cited papers. We carefully considered the reviewer's comments, and replaced "growth" by "linear extension", "budding" and "calcification" accordingly. We were also suggested to add comparison of the impact of pH on our coral recruits to *in situ* adults' colonies from the $CO_2$ vent system. According to this comment we emphasized the link of physiological responses from the laboratory and *in situ*.

Reviewer #2 main request was to highlight differences of responses to ocean warming and acidification between temperate and tropical coral early life stages. Although very few studies have been conducted on temperate species, we were able to add some comparison between our results and tropical species.

We believe that the changes made have greatly improved the manuscript and hope that it is now suitable for publication in *Biogeosciences*.

Yours sincerely,

Chloe Carbonne

On behalf of all co-authors

**Reviewer 1**

We thank reviewer #1 for his/her assessment and are pleased that s/he found the manuscript excellent on all criteria. We reply to his/her comments below. The comments are in bold while our reply is in plain font. Line numbers mentioned below refer to the revised manuscript.

- **"growth" may refer to many processes, please always specify whether you mean linear extension, or calcification (net or gross?), or polyp budding**
- **I have one comment left to improve clarity. As I mentioned in the initial submission comments, growth is a too general term and can refer to many aspects of coral biology. "Skeletal "growth" is commonly defined by the equation "net calcification rate = linear extension rate * skeletal bulk density". The whole three parameters make up "skeletal growth", and since they may display very different response to the same environmental variation, it should always be specified what particular aspect of growth the authors are referring to. Also make sure to be clear whether you refer to skeletal growth or to the production of new polyps in a colony, This applies widely to the introduction and discussion.**

As suggested, "growth" has been replaced by (1) "Linear extension" when speaking the extension of the surface of the polyp base is discussed (lines 18, 81, 215, 251, 272, 276, 285, 305, 320) and (2) "budding" to describe the formation of new polyp (lines 18, 81, 225, 252). "Growth" has also been replaced by "calcification" when appropriate (lines 307, 308).

- **add scale bar to each figure of a polyp/skeleton**

In the revised manuscript, scale bars have been added to Figure 4.A. and in the supplementary materials in Figure S2, S3, S5 and S6.

- **Abundance and polyp number per colony of this species were also measured in the field under OA, and the results of this laboratory study should consider those findings which mainly seem in agreement**.

Yes. This is right. Teixido et al (2020) measured abundance and polyp number per colony in the field at the CO2 vent system and reference sites with ambient pH. The authors found that *Astroides calycularis* was abundant (around 50 % of cover at 1 m depth) and colonies had fewer polyps at the CO2 vent compared to the colonies from reference areas. In the present study, only the recruits in the warm temperature – ambient pH treatment exhibited the highest number of polyps per colony with a mean of 1.33 polyps per recruit after 9 months. However, this result was not significative. We think that the overall low number of polyps found in this study may make difficult a clear link with the natural population structure in the field. However, following the reviewers' suggestion, we enlarged the concept of polyp' tissue retraction and skeletal density (see below) due to low pH in laboratory conditions and in the field. Polyp' tissue retraction found during our laboratory study may be linked with the reduction of coenosarc tissue found in the colonies at low pH in the field. These similar patterns of tissue reduction may be a common response to low pH with implications for calcification.

> Lines 282-287: "This phenomenon of being more predisposed to dissolution may be compared to the loss of coenosarc (the living tissue connecting the polyps) in adult colonies of *A. calycularis* that naturally occur in a $CO_2$ vent site in Ischia, where seawater is naturally acidified. (Teixido et al., 2020). As a response to low pH, colonies in the $CO_2$ vent showed a reduction of coenosarc compared to ambient pH sites. Interestingly, while the linear extension and structure of the skeleton were

affected by low pH, the density of the skeleton was higher. These results are in accordance with the higher density observed on adult colonies living in the $CO_2$ vent site in Ischia (Teixido et al., 2020)."

- **Also I recommend a spell checking, e.g. Carroselli at line 70 is misspelled.**

This misspelling has been corrected and the spelling of all authors has been checked.

**Reviewer 2**

**This manuscript describing and discussing the effects of thermal stress and acidification on larvae and juveniles of the mediterranean coral Astroides calycularis is mature and well written.**

We thank the reviewer for his/her kind words and reply to his/her comments below. The comments are in bold while our reply is in plain font. Line numbers mentioned below refer to the revised manuscript.

**I am very close to "accept as is" but I see two very minor points that could deserve some attention.**

- **At l 36, you refer to Garrabou et al. about "temperate corals". My understanding is that a large part of the database from Garrabou et al, is likely gorgonians (and actually Garrabou et al. refers to a Cnidaria category). I may be wrong, but if it is the case, I would recommend to clarify which "corals" are in this group as it can be easy to interpret "corals" as scleractinian corals. So a little clarification may be welcome.**

We agree with the referee. We have clarified the relevance of Garrabou et al. for our study. We also replaced this reference by their recently published article from the same first author that is more detailed.

> Lines 34-36: "Mediterranean anthozoans, including scleractinian corals (6 species) and gorgonians (7 species), have been suffering regular mass mortalities due to the increased intensity, duration and frequency of marine heatwaves (Garrabou et al., 2022)."

- **Also, and it is not absolutely necessary, but I am missing a few words about how Mediterranean/temperate ecosystems are distinct from the tropical corals ecosystems. Without doubts, some of the mechanisms can be generalised to both tropical and temperate ecosystems, but in addition to these, I would be curious to know if some stresses (or future scenarios) are more affecting temperate or tropical ecosystems. I believe these ecosystems are very different, so I would expect some unique responses too. (maybe one or two sentences in the introduction and discussion could be an idea.**

Too few studies have been conducted with temperate corals in early life stages in order to generalize a difference between tropical and temperate species. In the discussion part, the results of this study have been compared with the tropical coral studies and the trend of the responses are equivalent. A sentence has been completed at the end of the Discussion in order to underline that our temperate coral has, in general, similar responses to tropical corals early life stages.

> Lines 320-321: "Temperature acts on larval development while pH acts on the linear growth and calcification of the recruits as observed in previous studies on tropical species (e.g. Albright and Langdon 2011, Chua et al., 2013)."

However, under the double warming and acidification treatment a difference has been observed between *A. calycularis* and the tropical species which has been emphasized.

Lines 292-298: "The proportion of settlement of *A. calycularis* larvae was significantly lower in the warm temperature and low pH treatment. In contrast, other studies reported no effect of such treatment (e.g. Anlauf et al., 2011, Foster et al., 2015). This difference can be explained by the higher magnitude of the warming and acidification treatments used in the present study compared with other studies (+ 3 vs + 1-2°C and -0.3 vs -0.2 pH) (Anlauf et al., 2011, Chua et al., 2013, Bahr et al., 2020). Furthermore, all previous studies focused on tropical zooxanthellate corals which obtain additional energy by photosynthetic products translocated from their symbionts (Davy et al. 2012) as well as are subject to relatively uniform seawater temperatures (around 25 to 29°C), which differs with the high seasonal temperature variability of the Mediterranean (14°C in winter to 26°C in summer in Ischia)."

Regarding adult coral, a sentence in the Introduction compares tropical and Mediterranean response to low pH.

Lines 37-38: "However, temperate corals seem to be more tolerant to ocean acidification than tropical corals as their calcification is rarely affected by low pH (Rodolfo-Metalpa et al., 2011, Carbonne et al., 2021)."

- **Finally, and this is rather a proofreading type of comment, at l. 81, please add a space in A. calycularis.**

Done.

**It was a pleasure to review this manuscript at this mature stage and I look forward to see it published.**

Thank you!